## PERSPECTIVE

# The complex nature of blood pressure control during rest and exercise in humans: The role of the carotid chemoreflex

Jason H. Mateika[1,2,3] 🆔

[1]*John D. Dingell Veterans Affairs Medical Center, Detroit, MI, USA*

[2]*Department of Physiology, Wayne State University School of Medicine, Detroit, MI, USA*

[3]*Department of Internal Medicine, Wayne State University School of Medicine, Detroit, MI, USA*

Email: am1819@wayne.edu

Handling Editors: Harold Schultz & Andrew Holmes

The peer review history is available in the Supporting Information section of this article (https://doi.org/10.1113/JP288705#support-information-section).

Hypertension is a preventable medical condition that impacts ∼30% of the global adult population (Zhou et al., 2021). Under conditions of resting wakefulness, the blood pressure of around half the patients receiving treatment for hypertension remains uncontrolled (Zhou et al., 2021). Moreover, patients whose blood pressure is controlled under resting conditions often experience uncontrolled surges in blood pressure during exercise (Jones et al., 2022) that have been linked to increased risk of end-organ damage. In the present edition of *The Journal of Physiology*, Hinton et al. (2025) explore the role that the carotid chemoreflex has in blood pressure control during rest and exercise (i.e. submaximal exercise at 40–50% of maximal oxygen consumption) in a group of young (age = 28 ± 5 years) patients ($n = 8$ males and $n = 6$ females) with early-onset hypertension that was untreated. The hypertensive group was matched to a control group based on sex, age and height. Body mass index was higher in the hypertensive group; however, the body mass index of both groups was indicative of normal weight. To determine whether carotid chemoreflex sensitivity was higher in hypertensive individuals during rest and exercise, participants were exposed to a maximum of six episodes of hypoxia induced by the inspiration of 100% nitrogen interspersed with 3 min normoxic recovery periods. The duration of each hypoxic episode was varied to induce a range of mild to severe hypoxic levels across episodes. The ventilatory response to hypoxia was considered to be a non-invasive measure of carotid chemoreflex sensitivity.

The ventilatory response to hypoxia was measured after the infusion of saline and dopamine (i.e. an agent known to inhibit carotid chemoreflex discharge) that was administered randomly. The results showed that the carotid chemoreflex sensitivity was similar in the hypertensive and normotensive groups during rest and that the anticipated increase in sensitivity during exercise was similar in both groups. These findings were contrary to the hypothesis that chemoreflex sensitivity would be higher in the hypertensive group during both rest and exercise (Hinton et al., 2025). The results also showed that the administration of dopamine resulted in a reduction in the ventilatory response to hypoxia compared to measures after saline administration. However, a similar reduction was evident in the hypertensive and normotensive groups. In association with these findings, Hinton et al. (2025) also report that the blood pressure response to hypoxia during rest was reduced following dopamine infusion compared to saline. However, the reduction was similar in the hypertensive and normotensive groups. Moreover, reductions in blood pressure measures during rest and exercise under normoxic conditions were similar in the hypertensive and normotensive groups following dopamine infusion. One exception was noted: a reduction in systolic but not diastolic blood pressure was higher in the hypertensive group during rest following dopamine infusion.

Overall, contrary to the hypotheses of Hinton et al. (2025), increases in carotid chemoreflex sensitivity did not appear to have a prominent role in the hypertension that was evident in young untreated participants. On the one hand, the finding is robust because the strategy to recruit untreated hypertensive young patients ensured that potential confounding factors related to anti-hypertensive therapy were eliminated. Similarly, the strategy to match the hypertensive and normotensive groups based on sex, age and weight eliminated a number of confounding anthropometric factors that could be responsible for any differences between groups that were evident in the measured outcomes associated with the experimental design. On the other hand, as a consequence of the recruitment strategy, an exclusive group of hypertensive individuals were targeted and, consequently, the results of the study might not be applicable to other subpopulations of individuals living with hypertension. Likewise, the stringency of the matching strategy may have been partly responsible for the small *n* values that comprised both groups, which could have a role in reducing the power of some of the statistically non-significant findings that were reported.

As is typically the case with all well designed studies, many additional questions remain to be answered based on the results of the study. These additional questions include whether or not increased carotid chemoreflex sensitivity is an endotypic mechanism that has a predominant role in the development of hypertension in some patients and not others (i.e. young untreated hypertensive patients). Indeed, individuals with sleep apnoea that is induced predominantly by high loop gain (n.b. increased chemoreflex sensitivity contributes to high loop gain) (Puri et al., 2021) might be a population in which increased carotid chemoreflex sensitivity has a predominate role in the development of co-morbid hypertension. In addition, given the number of potential mechanisms (i.e. baroreflex, carotid chemoreflex and metaboreflex, to name but a few) that contribute to high blood pressure during rest and/or exercise, the manner in which these mechanisms interact and, ultimately, how this interaction might influence the pharmacological treatment of high blood pressure in different subpopulations of patients with hypertension is of interest. Lastly, previous work by Panza et al. (2022) has shown that the method used to measure the ventilatory response to hypoxia (i.e. intermittent episodes of hypoxia) is accompanied by a gradual increase in blood pressure that is sustained for at least 30 min following exposure. It would be of interest to determine whether the progressive increase in blood pressure is greater in hypertensive individuals and whether the administration of dopamine immediately after exposure results in a higher reduction in blood

pressure in hypertensive individuals under these experimental conditions.

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

## Additional information

### Competing interests

No competing interests declared.

### Author contributions

J.M. was responsible for the conception or design of the work; drafting the work or revising it critically for important intellectual content; and approving the final version of the manuscript submitted for publication. J.M. agrees to be accountable for all aspects of the work.

### Funding

This work was supported by the National Institutes of Health (R01HL142757) and the United States Department of Veterans Affairs (I01CX000125, I0BX006288, IK6CX002287).

### Keywords

carotid chemoreflex, exercise, humans, hypoxia

### Supporting information

Additional supporting information can be found online in the Supporting Information section at the end of the HTML view of the article. Supporting information files available:

**Peer Review History**

