## [Peer Review History · The Journal of Physiology]

The complex nature of blood pressure control during rest and exercise in humans; the role of the carotid chemoreflex

Jason H Mateika

DOI: 10.1113/JP288705

Corresponding author(s): Jason Mateika (jmateika@med.wayne.edu)

Review Timeline:

Submission Date:	20-Feb-2025
Editorial Decision:	04-Mar-2025
Revision Received:	09-Mar-2025
Accepted:	24-Mar-2025

Senior Editor: Harold Schultz

Reviewing Editor: Andrew Holmes

Transaction Report:

Dear Dr Mateika,

Re: JP-P-2025-288705 "**The complex nature of blood pressure control during restand exercise in humans; the role of the carotid chemoreflex** " by Jason H Mateika

Thank you for submitting your manuscript to The Journal of Physiology. It has been assessed by a Reviewing Editor and by 1 expert referee and we are pleased to tell you that it is acceptable for publication following satisfactory revision.

The review comments are copied at the end of this email.

Please address all the points raised and incorporate all requested revisions or explain in your Response to Referees why a change has not been made. We hope you will find the comments helpful and that you will be able to return your revised manuscript within 2 weeks. If you require longer than this, please contact journal staff: jp@physoc.org.

REVISION CHECKLIST:

We look forward to receiving your revised submission.

Yours sincerely,

Harold Schultz
Senior Editor
The Journal of Physiology

EDITOR COMMENTS

Reviewing Editor:

Comments to the Author:

Thank you for taking the time to write this perspectives article, we appreciate the effort that goes into such work. The piece nicely discusses the associated research article and does an excellent job at highlighting important questions that still need to be addressed moving forward. There are a couple of points raised by the reviewer that need some thought and I would appreciate your responses to these.

Senior Editor:

Comments to the Author:

Thank you for submitting this well written invited perspective article for consideration to the Journal of Physiology. The article has been reviewed by the focus authors who have a couple of minor comments. The authors' clarifications may or may not alter your position, but should be addressed. Please submit the article with your response.

REFEREE COMMENTS

Referee #1:

Thank you for your interest and review of the study. We really appreciate the effort put into reviewing the manuscript and writing this perspective.

The perspective brings up some important points about which specific populations the carotid chemoreflex might be important in driving some of the disease and outcomes for these patients. We agree that potentially patients with sleep apnoea (with high loop gain) and hypertension might have more of a chemoreflex role in their disease versus a young person with early onset hypertension.

We have several comments:

1. The author of the perspective points out that our application might be limited because we haven't chosen a broader group of patients with hypertension/or a representative group of hypertensive patients. However we disagree. Our patient group represents young adults with hypertension. The majority of patients with young onset hypertension are not put straight onto anti-hypertensive treatment. As per NICE guidelines (UK guidelines) there are lifestyle interventions first. Depending on the extent of organ damage, then some patients will be started on anti-hypertensives. We are concerned about young onset hypertension because these patients may have earlier onset cardiovascular events such as a stroke or MI. Thus we wanted to take steps to understand what mechanisms might contribute to their hypertension onset? Is it similar to those that have hypertension onset in mid-age, for example?

2. We appreciate the final point about using intermittent hypoxia and elevations in SBP at the end of the intermittent period. The author poses whether we saw a larger reduction in BP at rest with dopamine because we started with a higher BP after the intermittent hypoxia. Since we counterbalanced whether dopamine or saline was infused first (in a random order) - then this should remove any potential effect of elevated starting BP and a larger decrease in SBP with dopamine. This is if we have understood what the author posing. Apologies if we have misinterpreted this. If the author is interested we could examine the recover SBP after intermittent hypoxia? We do not remember observing any elevations in SBP as we usually wait for SBP, HR and ventilation to return to baseline before starting the next infusion.

END OF COMMENTS

1. The author of the perspective points out that our application might be limited because we haven't chosen a broader group of patients with hypertension/or a representative group of hypertensive patients. However, we disagree. Our patient group represents young adults with hypertension. The majority of patients with young onset hypertension are not put straight onto anti-hypertensive treatment. As per NICE guidelines (UK guidelines) there are lifestyle interventions first. Depending on the extent of organ damage, then some patients will be started on anti-hypertensives. We are concerned about young onset hypertension because these patients may have earlier onset cardiovascular events such as a stroke or MI. Thus we wanted to take steps to understand what mechanisms might contribute to their hypertension onset? Is it similar to those that have hypertension onset in mid-age, for example?

Whether or not the findings from this study are applicable to middle age or elderly individuals with high blood pressure accompanied by other co-morbidities (e.g. obesity, obstructive sleep apnea etc.) that could be accompanied by heightened carotid chemoreflex sensitivity that contributes to the elevated blood pressure remains to be determined. It is reasonable to suggest that additional studies on a wider population is necessary to answer this question, rather than suggest the findings will be applicable to all individuals with hypertension.

2. We appreciate the final point about using intermittent hypoxia and elevations in SBP at the end of the intermittent period. The author poses whether we saw a larger reduction in BP at rest with dopamine because we started with a higher BP after the intermittent hypoxia. Since we counterbalanced whether dopamine or saline was infused first (in a random order) - then this should remove any potential effect of elevated starting BP and a larger decrease in SBP with dopamine. This is if we have understood what the author posing. Apologies if we have misinterpreted this. If the author is interested we could examine the recover SBP after intermittent hypoxia? We do not remember observing any elevations in SBP as we usually wait for SBP, HR and ventilation to return to baseline before starting the next infusion.

My comment was unrelated to potential differences in starting blood pressure values prior to infusion of dopamine or saline (although the authors make a good point). The comment was made solely as an impetus to initiate studies to explore potential mechanisms that might be responsible for intermittent hypoxia induced long-term facilitation (i.e. sustained elevations) of blood pressure. For example, two groups (experimental vs. control) might be exposed to a well-established intermittent hypoxia protocol controlled for episode number, duration and intensity. Following exposure an immediate infusion of dopamine or saline might be administered to determine if long-term facilitation of the carotid chemoreflex is responsible for the manifestation of LTF of blood pressure.

Dear Dr Mateika,

Re: JP-P-2025-288705R1 "**The complex nature of blood pressure control during restand exercise in humans; the role of the carotid chemoreflex** " by Jason H Mateika

We are pleased to tell you that your paper has been accepted for publication in The Journal of Physiology.

Yours sincerely,

Harold Schultz
Senior Editor
The Journal of Physiology

If you would like to receive our 'Research Roundup', a monthly newsletter highlighting the cutting-edge research published in The Physiological Society's family of journals (The Journal of Physiology, Experimental Physiology, Physiological Reports, The Journal of Nutritional Physiology, and The Journal of Precision Medicine: Health and Disease), please click this link, fill in your name and email address and select 'Research Roundup':

<https://www.physoc.org/journals-and-media/membernews>

- You can help your research get the attention it deserves! Check out Wiley's free Promotion Guide for best-practice recommendations for promoting your work at: www.wileyauthors.com/eeo/guide. You can learn more about Wiley Editing Services which offers professional video, design, and writing services to create shareable video abstracts, infographics, conference posters, lay summaries, and research news stories for your research at: www.wileyauthors.com/eeo/promotion.

The Corresponding Author will receive an email from Wiley with details on how to register or log-in to Wiley Authors Services where you will be able to place an order

EDITOR COMMENTS

Reviewing Editor:

Comments to the Author:

Thank you again for putting together such a lively and thoughtful perspectives article. This is very much appreciated.

Senior Editor:

Comments to the Author:

The editors wish to thank the author for these final adjustments to the manuscript. The article is now accepted for publication. Congratulations for providing an interesting and insightful perspective article. Please consider the Journal of Physiology for your future works.

REFeree COMMENTS

Referee #1:

Thank you for responding. We have no further comments. The points made are reasonable.